# Novel Class Discovery for Point Cloud Segmentation via Joint Learning of Causal Representation and Reasoning

**Yang Li[1], Aming Wu[2], Zihao Zhang[1], Yahong Han[1]***

[1]School of Artificial Intelligence, College of Intelligence and Computing, Tianjin University, China
[2]School of Computer Science and Information Engineering, Hefei University of Technology, China
`liyang1389@tju.edu.cn, zhangzihao2490@tju.edu.cn, amwu@hfut.edu.cn, yahong@tju.edu.cn`

## Abstract

In this paper, we focus on Novel Class Discovery for Point Cloud Segmentation (3D-NCD), aiming to learn a model that can segment unlabeled (novel) 3D classes using only the supervision from labeled (base) 3D classes. The key to this task is to setup the exact correlations between the point representations and their base class labels, as well as the representation correlations between the points from base and novel classes. A coarse or statistical correlation learning may lead to the confusion in novel class inference. If we impose a causal relationship as a strong correlated constraint upon the learning process, the essential point cloud representations that accurately correspond to the classes should be uncovered. To this end, we introduce a structural causal model (SCM) to re-formalize the 3D-NCD problem and propose a new method, i.e., Joint Learning of Causal Representation and Reasoning. Specifically, we first analyze hidden confounders in the base class representations and the causal relationships between the base and novel classes through SCM. We devise a causal representation prototype that eliminates confounders to capture the causal representations of base classes. A graph structure is then used to model the causal relationships between the base classes' causal representation prototypes and the novel class prototypes, enabling causal reasoning from base to novel classes. Extensive experiments and visualization results on 3D and 2D NCD semantic segmentation demonstrate the superiorities of our method.

## 1 Introduction

Point Cloud Semantic Segmentation is one of the key tasks in autonomous driving Landrieu and Simonovsky (2017) and robotic perception Ghosh et al. (2017). However, traditional approaches adopt a "closed-world" assumption, which limits their applicability in real-world scenarios. The task of novel class discovery in point cloud semantic segmentation aims to learn a model that can segment unlabeled (novel) 3D classes using only the supervision from labeled (base) 3D classes Riz et al. (2023, 2024); Xu et al. (2024). The key to this task is to setup the exact correlations between the point representations and their base class labels, as well as the representation correlations between the points from base and novel classes. However, in practical scenarios, the base class classifier often learns shortcut features from the point cloud Geirhos et al. (2020); Hermann and Lampinen (2020), which are spurious correlations that hinder the learning of unbiased semantic representations for the base class, thereby easily misleading the model into making wrong predictions. Furthermore, due to the absence of labels and clear semantic connections with the base class, the novel class is prone to being misclassified as a base class. To this end, we analyze the problems as follows:

---

*Corresponding author.

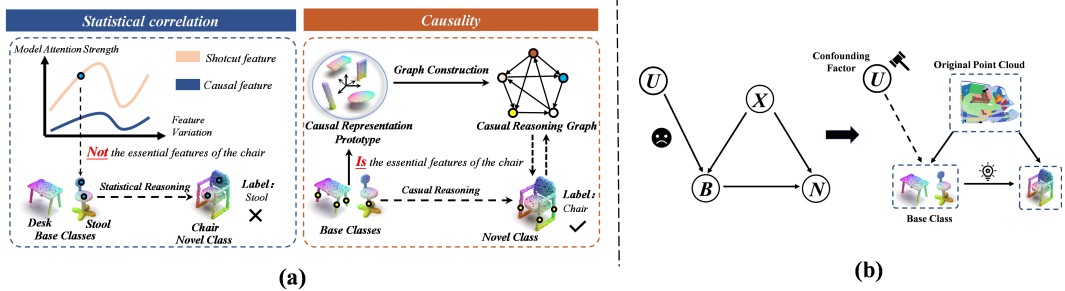

Figure 1: (a) On the left, misled by the shortcut feature of 'circular support', learning with statistical correlation fails to identify the novel class 'chair' but assign the label 'stool' straightforwardly. On the right, causality-based learning captures the base classes' causal representation (leg) and uses it as a key element for causal reasoning, correctly predicting the novel class. (b) SCM of 3D NCD: $B$ is the base class, $N$ the novel class, $X$ the original point cloud, and $U$ the confounding factor.

Firstly, traditional base class classifiers are essentially statistical models that focus on superficial dependencies between data and labels, without uncovering the intrinsic causal mechanisms. As a result, they tend to learn shortcut features, which are regarded as confounding factors in the field of causal learning Schölkopf et al. (2021). As shown in Figure 1 (a), statistical correlation learning may cause confusion in novel class inference. If we impose a causal relationship as a strongly correlated constraint during the learning process, the essential point cloud representations that accurately correspond to the classes should be uncovered.

Secondly, novel classes are often variations of base class through some causal mechanisms. For example, in point clouds, the posture of a 'rider' can be viewed as a causal variation of 'person' under the influence of the 'bicycle' context. Similarly, the structural similarity between 'truck' and 'car' reflects shared causal prior knowledge of 'vehicle.' Therefore, we need to leverage the causal relationships between base and novel classes to enhance the model's ability to infer novel classes.

To this end, we introduce a structural causal model (SCM) Pearl et al. (2016) to re-formalize the 3D-NCD problem. As shown in Figure 1 (b) , This SCM consists of four causal variables: the base class $B$ and novel class $N$ extracted from the original point cloud data $X$, the confounding factors $U$ (representing shortcut features or non-causal features). The path $U \rightarrow B$ indicates that the confounding factor affects the point cloud representation learning of the base class, while the path $B \rightarrow N$ reflects the causal impact of the base class on the novel class. Our goal is to remove $U$ to obtain the causal point cloud representation of base class and establish the causal relationship between the base and novel classes, thus enabling causal reasoning from $B$ to $N$.

To achieve this, we first introduce causal representation prototype learning, which eliminates confounding factors through causal adversarial mechanisms and extracts causal representations to obtain base class prototypes. Next, to enable causal reasoning, we introduce a graph structure with nodes representing the prototypes of the base and novel classes, and edges denoting their causal relationships. We then apply two constraints: causal pruning, which removes edges with low causal correlations to reduce irrelevant base class features on novel class learning, and inference direction consistency, which ensures the information flow in the graph aligns with causal relationships. Finally, the optimized graph is input into a graph convolutional network Kipf and Welling (2016) to generate base class labels and novel class pseudo-labels.

Our contribution can be summarized in three folds. *First*, we introduce a causality-based approach to excavate intrinsic causal mechanisms. This is the first method to incorporate causality into 3D NCD. *Second*, we propose causal representation prototype learning and graph-based causal reasoning methods to learn base classes' causal representation and model the causal relationships between base and novel classes. *Third*, extensive experiments and visualization results on 3D and 2D NCD semantic segmentation demonstrate the superiorities of our method.

## 2 Related work

**Point Cloud Semantic Segmentation.** The goal of 3D point cloud semantic segmentation is to partition a point cloud scene into different meaningful semantic parts. Traditional 3D semantic

segmentation methods Charles et al. (2017); Qi et al. (2017); Milioto et al. (2019) rely on full annotations, segmenting classes labeled in the training set. Several task settings extends the traditional 3D segmentation to the novel class. Among these, few-shot 3D segmentation An et al. (2025, 2024) reduces annotation burden but still requires limited labeled samples for novel classes. Similarly, zero-shot Liu et al. (2024a); Yang et al. (2023) and open-vocabulary Wei et al. (2025); Liu et al. (2024b) 3D segmentation rely on external semantic information like textual descriptions to identify novel classes. Novel class discovery Riz et al. (2023, 2024); Xu et al. (2024) in 3D segmentation aims to learn a model that can segment novel classes using only the supervision from labeled known classes. Critically, 3D-NCD demands neither the novel class samples of few-shot learning nor the textual guidance required by zero-shot and open-vocabulary methods. Instead, 3D-NCD autonomously discovers and segments novel classes by identifying patterns distinct from the known classes, offering a more practical and unconstrained solution for dynamic open-world environments (e.g., autonomous driving and domestic robotics), where objects may suddenly appear without prior definition or labels.

**Novel Class Discovery.** NCD leverages known classes to infer the semantics of novel classes. The 2D-NCD primarily follows two approaches. The first approach Hsu et al. (2018); Yen-Chang Hsu (2019) trains a classification network on known data and then clusters novel data based on the network's predictions. However, this method often overfits to the base classes, thereby limiting its performance on novel classes. The second approach Han et al. (2021); Wang et al. (2020a) incorporates both known and novel samples into clustering to promote shared feature representations, thus improving generalization. Although NCD has made some progress in 2D, research on NCD in the domain of 3D remains limited Weng et al. (2023); Riz et al. (2023, 2024); Xu et al. (2024). Additionally, these 3D-NCD methods rely on statistical similarity and overlook the causal relationships between base-novel classes. To address this, we introduce causal representation learning Schölkopf et al. (2021); Zhang et al. (2023) to capture the causal representation of base classes and model the causal relationships from base to novel classes, using a graphical structure to implement causal reasoning.

**Causal Learning in Computer Vision.** Causal relationships have recently been widely applied in learning-based 2D computer vision tasks Liu et al. (2024c); Zhang et al. (2023); Chen et al. (2023). Incorporating causality into machine learning helps to produce more learnable and interpretable models, since traditional CNN architectures consider only statistical correlation without accounting for causal structure. In the 3D point cloud domain, previous work such as CausalPC Huang et al. (2024) constructs a SCM Pearl et al. (2016), treating adversarial perturbations and sensor noise as the confounding factor $U$, and purifies point-cloud inputs by identifying and removing its causal effect to maintain classifier accuracy under various attacks. In contrast, in our method $U$ denotes non-causal shortcut features Geirhos et al. (2020); Hermann and Lampinen (2020) in the point cloud, which are eliminated via an adversarial network to enable novel class discovery and precise segmentation.

# 3 Method

**Problem Definition.** In point cloud segmentation, the NCD problem aims to train a segmentation network to process 3D datasets that contain partially labeled points from known classes and unlabeled points from novel classes, enabling the network to classify the 3D points in the scene into known or novel semantic classes. Formally, the training set consists of multiple 3D scenes, each containing two parts: 1) Labeled part: $D_s = \{(p_s^{(i)}, l_s^{(i)})\}_{i=1}^H$, where $p_s^{(i)}$ represents the $i$-th point, and $l_s^{(i)} \in A_s$ is its corresponding known class label. 2) Unlabeled part: $D_u = \{p_u^{(j)}\}_{j=1}^L$, where $p_u^{(j)}$ represents the $j$-th point belonging to the novel class set $A_u$, and $A_s \cap A_u = \varnothing$ (the known class set and the novel class set do not overlap). The goal is to train a point cloud segmentation network $F$ that can accurately classify each point in novel scenes from the test set into one of the classes in $A_s \cup A_u$.

## 3.1 A Causal View of Novel Class Discovery in Point Cloud Segmentation

We model this task using an SCM Pearl et al. (2016) shown in the Figure 1 (b). This SCM consists of four causal variables: the base class $B$ and novel class $N$ extracted from the original point cloud data $X$, the confounding factors $U$. We now present the three main causal paths in this task.

1) $B \leftarrow X \rightarrow N$: This represents the the base class point cloud data $B$ and novel class point cloud data $N$ extracted from the original point cloud data $X$, with a causal path between them.

2) $B \rightarrow N$: The base class $B$ influence the novel class $N$ through some causal mechanisms. For example, the features of 'person' are adjusted by the context of 'bicycle' to generate the features of 'rider'. The similarity between 'truck' and 'car' reflects shared causal prior knowledge of 'vehicle.'

3) $U \rightarrow B$: The model often learns shortcut features Geirhos et al. (2020) from the base class point cloud data $B$, which are regarded as confounding factors $U$ that hinder the learning of base classes' causal representations. As shown in Figure 1 (a), $U$ could be a visually prominent but non-causal element like a 'circular support', whereas the base classes' causal representation is the 'leg'.

## 3.2 Causal Representation Prototype Learning

We first use the feature extractor $f_\theta$ to extract the initial features $Z \in \mathbb{R}^{P \times d}$ for the base class and $Z' \in \mathbb{R}^{P \times d}$ for the novel class from the point cloud data $X$, where $P$ is the number of sampled points and $d$ is the feature dimension. We use MinkowskiUNet Choy et al. (2019) as the backbone. For the base class, we denote the point cloud as $X_B$, and $Y_B$ is the corresponding label. After clustering, we obtain the set of novel class prototypes $\{n_1, n_2, \ldots, n_K\}$, and at time $t = 0$, we define the base class causal representation prototypes as $C = \{c_1^{(0)}, c_2^{(0)}, \ldots, c_M^{(0)}\}$.

In Figure 1 (b), the causal path $U \rightarrow B$ leads to spurious correlations in the learned $P(B|Z)$ rather than the true causal relationship $P(B|\mathrm{do}(Z_{\mathrm{causal}}))$, where the $\mathrm{do}(\cdot)$ operator denotes *causal intervention* Pearl et al. (2016). Our core objective is to learn a pure causal representation $Z_c$ for the base class $B$, uncontaminated by the confounding factor $U$. Theoretically, if $U$ were observable, its influence could be eliminated via the backdoor adjustment formula Pearl et al. (2016): $P(B|\mathrm{do}(Z)) = \sum_u P(B|Z,u)P(u)$. However, confounding factors are often unknown or difficult to define explicitly. Existing representative 2D causality-based methods simplify them to the average of visual features Wang et al. (2021, 2020b), but this may not be accurate enough for complex point cloud data.

Therefore, we aim to learn a feature representation $Z$ that is mutually independent of the confounding factor $U$ (i.e., $Z \perp U$), which involves minimizing the mutual information $I(Z; U)$ between them. This objective aligns closely with the following fundamental causal principle:

**Principle 1 (Schoelkopf et al. (2012)).** *Independent Causal Mechanisms (ICM) Principle: The conditional distribution of each variable given its causes (i.e., its mechanism) does not inform or influence the other mechanisms.*

According to the *ICM* principle, the true causal mechanism generating the base class $B$ should be independent of the mechanism producing the confounder $U$. Striving for $Z \perp U$ helps to disentangle these mechanisms, enabling $Z$ to stably represent the intrinsic properties of $B$ without being perturbed by variations in $U$. Adversarial training provides an indirect way to handle implicit confounding factors. Drawing inspiration from GAN Goodfellow et al. (2014) , we design an adversarial process:

$$\min_\theta \max_\phi \mathcal{L}_{ADV} = \mathcal{L}_{cls}(f_\theta(X_B), Y_B) - \lambda_{adv}\mathcal{L}_{adv}(g_\phi(Z), U). \tag{1}$$

Here, $\lambda_{adv}$ is the weight coefficient of the adversarial loss, balancing the losses. The feature extractor $f_\theta$ aims to extract the causal features of base class data $X_B$ while minimizing components related to $U$, whereas the adversarial network $g_\phi$ attempts to recover the confounding factor $U$ from $Z$. If $g_\phi$ successfully recovers $U$, it indicates that $Z$ still contains confounding information; if not, $Z$ has effectively removed the interference from $U$. This process can be seen as a mechanism for approximating $Z \perp U$ and encouraging the learning of representations compliant with the ICM principle. $\mathcal{L}_{cls}$ is the classification loss, implemented with a cross-entropy loss function:

$$\mathcal{L}_{cls}(f_\theta(X_B), Y_B) = -\sum_i Y_i \log(f_\theta(X_B)_i). \tag{2}$$

$\mathcal{L}_{adv}$ is the adversarial loss, aiming to maximize the ability of the adversarial network $g_\phi$ to recover $U$ from $Z$, while minimizing the correlation between $Z$ and $U$. This is achieved using binary cross-entropy loss:

$$\mathcal{L}_{adv}(g_\phi(Z), U) = -\mathbb{E}[\log(g_\phi(Z) \cdot U + (1 - g_\phi(Z)) \cdot (1 - U))]. \tag{3}$$

After adversarial training, we update the prototypes $C$ from the optimized features $Z$. The normalized similarity weight between $j$-th point cloud feature and $i$-th prototype is:

$$W_{ij} = \frac{\exp(\mathrm{sim}(Z_j, C_i))}{\sum_{k=1}^{M} \exp(\mathrm{sim}(Z_j, C_k))}, \tag{4}$$

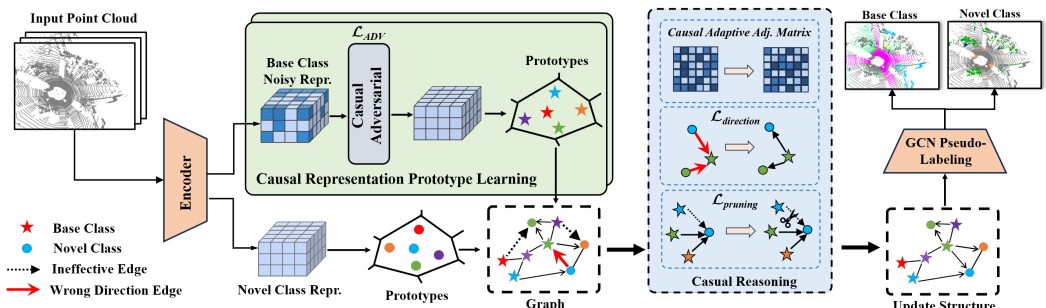

Figure 2: **Overview of our method.** We extract the base and novel class representations from the initial point cloud. In the base class point cloud data, there exist noisy confounding factors. We obtain causal representations through causal adversarial deconfounding and then generate base class prototypes. Meanwhile, the novel class representation also generates prototypes, which are combined with the base class causal representation prototypes as nodes to construct the graph. The causal adaptive adjacency matrix dynamically adjusts the graph structure, $\mathcal{L}_{\text{pruning}}$ aims to identify and remove ineffective edges, and $\mathcal{L}_{\text{direction}}$ ensures that the information propagation direction between nodes aligns with the actual causal path. After these causal reasoning optimizations, the updated graph is input into the GCN to generate labels for base classes and pseudo-labels for novel classes.

where $\text{sim}(Z_j, C_i)$ is the similarity between the point cloud feature $Z_j$ and the prototype $C_i$, implemented using cosine similarity. Here, $j \in \{1, 2, \ldots, P\}$ represents all sampled points in the point cloud. The updated value of the $i$-th prototype after the $t + 1$-th iteration is computed as:

$$C_i^{(t+1)} = \frac{\sum_j W_{ij} \cdot Z_j}{\sum_j W_{ij}}. \tag{5}$$

To ensure efficient and effective matching between the point cloud features $Z$ and the prototypes $C$, we use the causal prototype matching loss, which is computed as:

$$\mathcal{L}_{PRO} = -\sum_{i=1}^{M} \sum_{j=1}^{P} W_{ij} \cdot \text{sim}(Z_j, C_i) + \lambda \cdot \|C_i\|_2^2, \tag{6}$$

where $\lambda$ is the regularization coefficient that controls prototype complexity and prevent overfitting.

Please note that the extraction of causal representations relies on learning clear and reliable causal relationships from data. Although novel class features $Z'$ also contain confounding factors, due to the lack of explicit labels, no direct supervision can be provided. The pseudo-labels for the novel class relies on the model's assumptions and reasoning, which may introduce bias or noise, and we cannot ensure that the features learned from the pseudo-labels are consistent with the true causal relationships. Thus, this work focuses on extracting causal representations for base classes. During novel class inference, these representations serve as reliable prior knowledge, helping the model understand novel class features through causal reasoning.

### 3.3 Causal Reasoning Graph Construction

Figure 1 (b) illustrates the causal path from the base to the novel class ($B \rightarrow N$), and our key challenge is how to effectively model this causal path in a high-dimensional representation space. A natural idea is to impose a structured representation of the potential causal influences from base to novel classes, yielding a more robust and interpretable framework for knowledge transfer. To this end, we draw inspiration from Causal Bayesian Networks (CBNs) Pearl (2009), which use directed edges to unambiguously encode causal dependencies. Underlying CBNs is a fundamental assumption:

**Principle 2 (Spirtes et al. (1993)).** *Causal Markov Principle: In a causal DAG G, each variable is conditionally independent of all non-descendant variables given its direct parents.*

This aligns with the essence of NCD task: the representation of a novel class should be influenced by the base classes that are its direct causes, while also incorporating novel features inherent to the novel class and not fully explained by the bases. Based on the above thinking, we construct

a graph to explicitly model the hypothesized causal path ($B \rightarrow N$). The graph consists of causal representation prototypes of $M$ base classes, denoted as $C = \{c_1, c_2, \ldots, c_M\}$, and $K$ novel class prototypes, denoted as $N = \{n_1, n_2, \ldots, n_K\}$. The node set is $V = C \cup N$. The graph contains a set of edges $E_{\text{causal}} = \{(c_i, n_j) \mid 1 \leq i \leq M, 1 \leq j \leq K\}$, where $w_{ij}$ is the weight of the edge.

To address dynamic novel class inputs and the catastrophic forgetting problem in open world, we design a causal adaptive adjacency matrix $A = [A_{ij}]_{M \times K}$, where each element $A_{ij} = w_{ij}$ represents the causal relationship strength from the node $c_i$ to the node $n_j$. By introducing a self-attention mechanism Vaswani et al. (2017), the weight $w_{ij}$ of each edge is dynamically adjusted:

$$w_{ij} = \text{softmax}\left(\frac{\text{Attention}(c_i, n_j)}{\tau}\right), \tag{7}$$

where $\tau$ is the temperature coefficient. During information propagation in the set $E_{\text{causal}}$, the causal path $B \rightarrow N$ may not always be strictly followed, leading to ineffective causal knowledge transfer or interference. To ensure causal directionality, we introduce the inference direction consistency constraint:

$$\mathcal{L}_{\text{direction}} = \sum_{(c_i, n_j) \in E} \left(w_{ij} \cdot (1 - \mathbb{I}(c_i \rightarrow n_j))\right)^2. \tag{8}$$

$\mathbb{I}(c_i \rightarrow n_j)$ is an indicator function where $\mathbb{I}(c_i \rightarrow n_j) = 1$ indicates a correct causal path with zero loss. If the propagation direction is incorrect, the loss increases with the edge weight $w_{ij}$, guiding the model to learn the correct direction.

In the NCD task, many base class nodes may have a weak or irrelevant impact on the novel class nodes, leading to ineffective information transmission and interfering with the generation of pseudo-labels for the novel class. To mitigate this, we introduce a causal pruning constraint, removing edges with causal weights below a learnable threshold $\theta$:

$$\mathcal{L}_{\text{pruning}}(\theta) = \sum_{(c_i, n_j) \in E} \mathbb{I}(w_{ij} < \theta) \cdot w_{ij}^2. \tag{9}$$

Here, $\mathbb{I}(w_{ij} < \theta)$ indicates edge removal, while $w_{ij}^2$ penalizes weak connections. As causal relationships in the $E_{\text{causal}}$ change with novel class nodes input, the model dynamically adjusts the threshold $\theta$, which is set as a learnable parameter.

### 3.4 Pseudo-label Generation Based on GCN

Existing 3D NCD methods generate pseudo-labels by directly measuring similarity between novel and base classes, ignoring the complex higher-order dependencies between classes. To address this, we introduce a graph convolutional network, which can handle the complex relationships between novel and base classes by leveraging multi-layer propagation and neighbor aggregation to generate high-quality labels. The update rule for each novel class node $n_j$ is:

$$\mathbf{n}_j^{(t+1)} = \sigma\left(\sum_{i=1}^{M} \frac{w_{ij}}{\sqrt{d_i d_j}} \cdot \mathbf{c}_i^{(t)} + \sum_{k=1}^{K} \frac{w_{jk}}{\sqrt{d_j d_k}} \cdot \mathbf{n}_k^{(t)}\right). \tag{10}$$

Here, $d_i, d_j, d_k$ are node degrees, and $\sigma$ is LeakyReLU. After multiple GCN layers, the final representation of novel class node $\mathbf{n}_j^{\text{final}}$ is used to assign pseudo-labels:

$$\hat{y}_j = \arg\min_i \text{sim}(\mathbf{n}_j^{\text{final}}, \mathbf{c}_i). \tag{11}$$

Where $\hat{y}_j$ is the predicted pseudo-label for the novel class final representation and similarity is measured via cosine similarity.

## 4 Experiments

### 4.1 Experimental Setup

**Datasets and Evaluation Metrics.** We evaluated our method on the SemanticKITTI Behley et al. (2019) and SemanticPOSS Pan et al. (2020) datasets. We adopted the same dataset partitioning

Table 1: The novel class discovery results on SemanticPOSS dataset. 'Full' denotes the results obtained by supervised learning. The green values are the novel classes in each split.

| Split | Method | bike | build. | car | cone | fence | grou. | pers. | plants | pole | rider | traf. | trashc. | trunk | Novel | Known | All |
|---|---|---|---|---|---|---|---|---|---|---|---|---|---|---|---|---|---|
| | Full | 48.3 | 86.9 | 57.2 | 38.5 | 49.8 | 79.1 | 63.8 | 81.7 | 37.0 | 58.1 | 33.5 | 9.1 | 27.4 | - | - | 51.5 |
| 0 | EUMS Zhao et al. (2021) | 25.7 | 4.0 | 0.6 | 16.4 | 29.4 | 36.8 | 43.8 | 28.5 | 13.1 | 26.8 | 18.2 | 3.3 | 16.9 | 17.4 | 21.5 | 20.3 |
| | NOPS Riz et al. (2023) | 35.5 | 30.4 | 1.2 | 13.5 | 24.1 | 69.1 | 44.7 | 42.1 | 19.2 | 47.7 | 24.4 | 8.2 | 21.8 | 35.7 | 26.6 | 29.4 |
| | SNOPS Riz et al. (2024) | 34.2 | 58.8 | 10.0 | 13.2 | 18.7 | 77.3 | 45.8 | 58.6 | 17.3 | 48.4 | 22.6 | 8.7 | 22.9 | 51.2 | 25.8 | 33.6 |
| | DASL Xu et al. (2024) | 46.3 | 51.5 | 6.0 | 35.7 | 48.5 | 83.0 | 67.9 | 53.1 | 35.5 | 59.3 | 31.0 | 2.8 | 15.5 | 48.4 | 38.0 | 41.2 |
| | Ours | 46.1 | 60.4 | 8.3 | 40.4 | 51.1 | 80.8 | 67.1 | 55.3 | 39.1 | 58.2 | 19.4 | 2.1 | 13.3 | 51.3 | 37.4 | 41.7 |
| 1 | EUMS Zhao et al. (2021) | 15.2 | 68.0 | 28.0 | 24.0 | 11.9 | 75.1 | 36.0 | 74.5 | 26.9 | 48.6 | 26.0 | 5.6 | 23.1 | 21.0 | 40.0 | 35.6 |
| | NOPS Riz et al. (2023) | 29.4 | 71.4 | 28.7 | 12.2 | 3.9 | 78.2 | 56.8 | 74.2 | 18.3 | 38.9 | 23.3 | 13.7 | 23.5 | 30.0 | 38.2 | 36.4 |
| | SNOPS Riz et al. (2024) | 16.3 | 71.4 | 30.0 | 19.8 | 24.9 | 77.1 | 55.0 | 73.4 | 15.8 | 38.4 | 22.3 | 15.7 | 23.6 | 32.1 | 38.7 | 37.2 |
| | DASL Xu et al. (2024) | 31.5 | 83.2 | 48.7 | 25.4 | 23.9 | 77.3 | 53.1 | 77.1 | 32.5 | 57.3 | 35.0 | 9.3 | 18.0 | 36.2 | 46.4 | 44.0 |
| | Ours | 32.2 | 84.0 | 49.1 | 32.4 | 17.3 | 77.5 | 62.3 | 77.9 | 34.0 | 62.0 | 36.8 | 9.1 | 20.8 | 37.3 | 48.4 | 45.8 |
| 2 | EUMS Zhao et al. (2021) | 40.1 | 69.5 | 27.7 | 13.5 | 34.9 | 76.0 | 54.7 | 75.6 | 5.3 | 39.2 | 7.8 | 8.5 | 11.9 | 8.3 | 44.0 | 35.7 |
| | NOPS Riz et al. (2023) | 37.2 | 71.8 | 29.7 | 14.6 | 28.4 | 77.5 | 52.1 | 73.0 | 11.5 | 47.1 | 0.5 | 10.2 | 14.8 | 9.0 | 44.2 | 36.0 |
| | SNOPS Riz et al. (2024) | 38.4 | 72.5 | 28.0 | 14.5 | 26.2 | 78.1 | 54.7 | 74.3 | 10.0 | 48.3 | 23.0 | 10.2 | 17.7 | 16.9 | 44.5 | 38.1 |
| | DASL Xu et al. (2024) | 45.3 | 82.8 | 49.8 | 28.4 | 46.3 | 76.1 | 66.2 | 77.2 | 10.9 | 58.4 | 18.6 | 7.3 | 8.2 | 12.6 | 53.8 | 44.3 |
| | Ours | 45.1 | 83.3 | 52.6 | 38.0 | 46.9 | 78.9 | 68.6 | 78.7 | 19.8 | 59.4 | 14.8 | 9.0 | 6.2 | 13.6 | 56.1 | 46.3 |
| 3 | EUMS Zhao et al. (2021) | 41.2 | 70.7 | 28.1 | 4.3 | 38.3 | 76.7 | 38.3 | 75.4 | 25.8 | 34.3 | 28.3 | 0.4 | 24.4 | 13.0 | 44.7 | 37.4 |
| | NOPS Riz et al. (2023) | 38.6 | 70.4 | 30.9 | 0.0 | 29.4 | 76.5 | 56.0 | 71.8 | 17.0 | 31.9 | 36.3 | 1.0 | 22.6 | 10.9 | 43.9 | 36.3 |
| | SNOPS Riz et al. (2024) | 39.4 | 70.3 | 30.0 | 9.1 | 26.8 | 77.6 | 54.3 | 72.5 | 16.0 | 49.9 | 28.1 | 1.3 | 23.5 | 20.1 | 43.9 | 38.4 |
| | DASL Xu et al. (2024) | 45.5 | 82.9 | 47.7 | 0.0 | 45.1 | 77.8 | 66.3 | 77.7 | 34.3 | 49.1 | 30.5 | 4.0 | 15.3 | 17.7 | 52.8 | 44.7 |
| | Ours | 44.7 | 81.7 | 50.0 | 36.2 | 47.4 | 75.1 | 65.1 | 76.5 | 31.0 | 47.5 | 33.9 | 0.0 | 24.9 | 27.9 | 53.0 | 47.2 |

Table 2: The novel class discovery results on the SemanticKITTI dataset. 'Full' denotes the results obtained by supervised learning. The four groups represent the four splits in turn.

| Method | bi.cle | b.clst | build. | car | fence | mt.cle | m.clst | oth-g. | oth-v. | park. | pers. | pole | road | sidew. | terra. | traff. | truck | trunk | veget. | Novel | Known | All |
|---|---|---|---|---|---|---|---|---|---|---|---|---|---|---|---|---|---|---|---|---|---|---|
| Full | 4.8 | 59.2 | 87.1 | 92.5 | 36.5 | 28.0 | 2.5 | 4.0 | 27.8 | 39.1 | 35.4 | 63.4 | 90.8 | 77.1 | 63.7 | 41.5 | 55.0 | 58.1 | 90.1 | - | - | 50.3 |
| EUMS Zhao et al. (2021) | 5.3 | 40.0 | 15.8 | 79.2 | 9.0 | 16.9 | 2.5 | 0.1 | 11.4 | 14.4 | 12.7 | 29.2 | 42.6 | 26.1 | 0.1 | 10.3 | 47.4 | 37.9 | 38.4 | 24.6 | 21.1 | 23.1 |
| NOPS Riz et al. (2023) | 5.6 | 47.8 | 52.7 | 82.6 | 13.8 | 25.6 | 1.4 | 1.7 | 14.5 | 19.8 | 25.9 | 32.1 | 56.7 | 8.1 | 23.8 | 14.3 | 49.4 | 36.2 | 44.2 | 37.1 | 26.5 | 29.3 |
| SNOPS Riz et al. (2024) | 6.6 | 43.9 | 72.0 | 83.3 | 13.6 | 24.7 | 2.5 | 2.4 | 15.1 | 18.7 | 24.6 | 31.6 | 49.5 | 43.2 | 27.4 | 15.7 | 42.1 | 38.5 | 37.5 | 45.9 | 26.0 | 31.2 |
| DASL Xu et al. (2024) | 5.5 | 51.1 | 74.6 | 92.3 | 29.8 | 22.8 | 0.0 | 0.0 | 23.3 | 24.8 | 27.7 | 59.7 | 41.4 | 22.5 | 23.6 | 39.3 | 43.6 | 51.1 | 66.4 | 45.7 | 33.7 | 36.8 |
| Ours | 5.2 | 48.9 | 70.5 | 90.4 | 29.8 | 21.4 | 0.6 | 0.0 | 26.0 | 22.5 | 23.0 | 56.3 | 53.1 | 24.1 | 23.7 | 33.5 | 41.1 | 51.4 | 63.2 | 46.9 | 32.2 | 36.9 |
| EUMS Zhao et al. (2021) | 7.5 | 42.4 | 80.0 | 76.8 | 8.6 | 19.6 | 1.4 | 0.6 | 12.0 | 14.1 | 14.0 | 40.7 | 86.3 | 66.5 | 56.3 | 12.0 | 44.8 | 20.9 | 72.4 | 24.2 | 37.1 | 35.6 |
| NOPS Riz et al. (2023) | 7.4 | 51.2 | 84.5 | 50.9 | 7.3 | 28.9 | 1.8 | 0.0 | 22.2 | 19.4 | 30.4 | 37.6 | 90.1 | 72.2 | 60.8 | 16.8 | 57.3 | 49.3 | 85.1 | 25.4 | 46.2 | 40.7 |
| SNOPS Riz et al. (2024) | 7.6 | 43.5 | 85.1 | 68.7 | 18.9 | 24.4 | 3.5 | 0.0 | 23.9 | 19.1 | 27.0 | 36.5 | 89.3 | 71.9 | 62.0 | 17.2 | 55.9 | 29.4 | 84.4 | 27.2 | 45.2 | 40.4 |
| DASL Xu et al. (2024) | 3.7 | 57.4 | 89.2 | 56.5 | 17.3 | 20.3 | 0.0 | 0.0 | 20.0 | 30.6 | 34.8 | 60.6 | 93.2 | 77.6 | 62.0 | 38.7 | 56.9 | 39.2 | 86.7 | 28.7 | 50.1 | 44.5 |
| Ours | 4.3 | 51.9 | 89.4 | 82.6 | 13.9 | 25.5 | 0.0 | 0.0 | 25.2 | 28.7 | 34.7 | 62.0 | 92.6 | 77.0 | 62.7 | 37.5 | 62.3 | 42.8 | 87.6 | 33.6 | 50.9 | 46.6 |
| EUMS Zhao et al. (2021) | 8.3 | 50.8 | 83.0 | 88.1 | 17.9 | 2.8 | 2.3 | 0.2 | 3.2 | 25.4 | 25.0 | 20.2 | 88.3 | 71.0 | 57.9 | 8.6 | 27.2 | 38.4 | 77.0 | 12.4 | 42.2 | 36.6 |
| NOPS Riz et al. (2023) | 6.7 | 49.2 | 86.4 | 90.8 | 23.7 | 2.7 | 0.6 | 1.9 | 15.5 | 29.5 | 27.9 | 36.4 | 90.3 | 73.4 | 61.2 | 17.8 | 10.3 | 46.2 | 84.3 | 16.5 | 48.0 | 39.7 |
| SNOPS Riz et al. (2024) | 6.8 | 48.3 | 86.1 | 89.9 | 22.2 | 9.3 | 0.6 | 3.6 | 10.5 | 28.4 | 27.1 | 23.8 | 90.6 | 73.8 | 61.9 | 22.3 | 22.1 | 46.1 | 83.8 | 17.6 | 48.0 | 39.8 |
| DASL Xu et al. (2024) | 3.6 | 54.2 | 88.9 | 93.3 | 28.4 | 10.2 | 0.0 | 0.9 | 9.6 | 33.4 | 32.2 | 36.1 | 92.7 | 77.4 | 62.2 | 10.7 | 34.2 | 51.7 | 86.9 | 20.1 | 50.4 | 42.5 |
| Ours | 8.0 | 66.0 | 89.2 | 93.3 | 27.7 | 21.7 | 0.0 | 0.2 | 9.6 | 33.8 | 34.0 | 33.1 | 93.1 | 77.9 | 61.7 | 20.5 | 7.8 | 52.8 | 86.8 | 18.5 | 51.8 | 43.0 |
| EUMS Zhao et al. (2021) | 4.0 | 2.5 | 80.1 | 87.2 | 16.8 | 14.0 | 15.0 | 0.3 | 14.1 | 20.8 | 6.8 | 37.6 | 86.8 | 66.5 | 55.3 | 16.2 | 40.6 | 38.4 | 76.2 | 7.1 | 43.4 | 35.7 |
| NOPS Riz et al. (2023) | 2.3 | 27.8 | 86.0 | 89.9 | 23.1 | 24.5 | 2.9 | 3.1 | 18.2 | 30.1 | 16.3 | 39.9 | 90.7 | 73.5 | 61.0 | 17.4 | 49.8 | 44.0 | 83.2 | 12.4 | 49.0 | 41.2 |
| SNOPS Riz et al. (2024) | 4.7 | 31.5 | 84.6 | 88.7 | 22.8 | 23.3 | 8.2 | 2.6 | 17.9 | 28.7 | 15.1 | 38.3 | 89.7 | 72.5 | 60.8 | 16.1 | 43.3 | 45.7 | 82.9 | 14.9 | 47.9 | 40.9 |
| DASL Xu et al. (2024) | 2.6 | 32.5 | 88.7 | 93.3 | 28.1 | 24.0 | 0.1 | 1.0 | 23.7 | 35.6 | 15.3 | 59.8 | 93.2 | 77.6 | 61.4 | 37.8 | 56.6 | 52.1 | 86.7 | 12.6 | 54.6 | 45.8 |
| Ours | 0.5 | 40.0 | 89.4 | 93.5 | 29.7 | 24.6 | 6.2 | 0.2 | 24.8 | 36.4 | 13.8 | 61.5 | 93.5 | 77.8 | 62.6 | 40.0 | 60.9 | 52.9 | 87.4 | 15.1 | 55.7 | 46.8 |

strategy as NOPS Riz et al. (2023), where one subset is designated as the novel classes, and the remaining subsets as the base classes. We evaluated on sequences 08 and 03 from the SemanticKITTI and SemanticPOSS datasets, respectively, which include both known and novel classes. For the known classes, we report the IoU for each class. For the novel classes, we use the Hungarian algorithm Kuhn (2010) to match cluster labels with ground truth labels and then present the IoU values for each novel class.

**Implementation Details.** We implemented our network using the MinkowskiUNet-34C Choy et al. (2019) architecture, consistent with existing methods Riz et al. (2023, 2024); Xu et al. (2024). We extracted point-level features from the penultimate layer. The number of prototypes for both base and novel classes is consistent with the number of classes in the dataset. The optimizer used is AdamW, with an initial learning rate of 1e-3, decaying every 5 epochs until reaching a minimum value of 1e-5. The weight coefficient $\lambda_{adv}$, which balances classification and adversarial losses, and the threshold $\theta$ in the causal pruning constraint are both initially set to 0.5 and are dynamically adjusted during network training. For the hyperparameters, we set the temperature parameter $\tau$ to 0.06 and the regularization coefficient $\lambda$ to 0.02. The number of graph convolution layers is set to 3.

## 4.2 Comparison with the State of the Art

**SemanticPOSS dataset.** The results in Table 1 show that our method achieves the highest mIoU across all classes in all splits. In split 2, the base class mIoU reaches 56.1%, a 3.3% improvement over DASL, demonstrating effective mitigation of base class confounding through causal representation learning. In the challenging split 3, our method outperforms DASL on novel classes by 10.2%, with

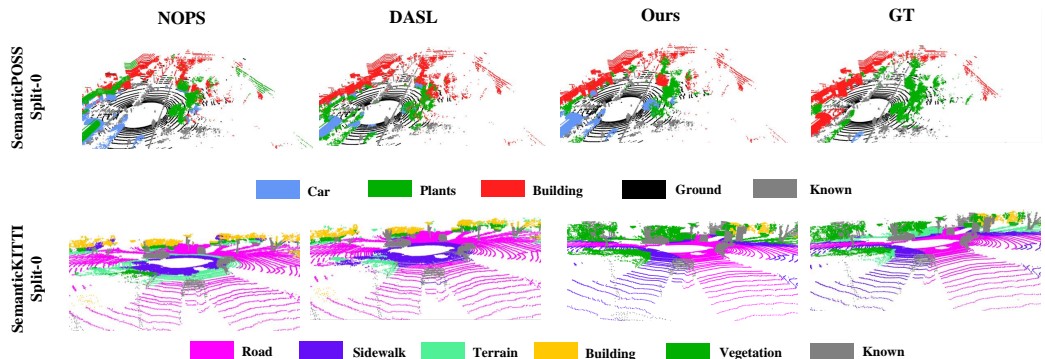

Figure 3: **Visualization comparison between our method, NOPS, and DASL on the SemanticPOSS and SemanticKITTI datasets.** In the first row, NOPS and DASL, relying on statistical methods, fail to eliminate non-causal features, causing confusion between 'Plants' and 'Building'. In the second row, they also confuse 'Road' with 'Sidewalk' and 'Terrain' with 'Vegetation'. In contrast, our method achieves better results and generates high-quality pseudo labels.

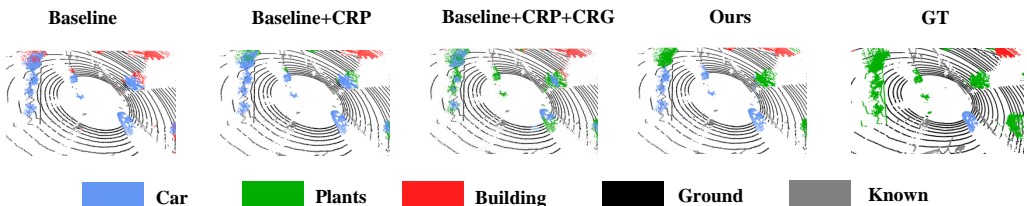

Figure 4: **Ablation experiment visualization.** The introduction of CRP significantly reduced the misclassification between 'Plant' and 'Building.' Subsequently, integrating CRG further alleviated the confusion between 'Plant' and 'Car.' Finally, incorporating GCPL effectively reduced the confusion among 'Plant,' 'Car,' and 'Building.' .

a notable 36.2% for the cone-stone class, while both DASL and NOPS achieve 0. These results demonstrate that our use of causal reasoning enhances the segmentation accuracy for novel classes.

**SemanticKITTI dataset.** The results in Table 2 show that our method achieves the highest mIoU across all classes in all splits. Our method improves novel class segmentation by 4.9% on split 1. Specifically, in the car class, it achieves an IoU of 82.6%, a 13.9% improvement over SNOPS. In the motorcycle class of split 3, our method achieves an 11.5% improvement. Figure 3 presents a visual comparison of NOPS, DASL, and our method.

### 4.3 Ablation Study

Our method consists of three components: causal representation prototype learning (CRP), causal reasoning graph (CRG), and graph convolutional based pseudo-label generation (GCPL). Table 3 presents the performance improvements at each stage. The first row shows the baseline model, a modified MinkowskiUNet-34C Choy et al. (2019). The second row demonstrates that introducing CRP enhances novel class segmentation by capturing causal features. In the third row, we replace the causal representation prototypes with prototypes generated by traditional clustering as graph

Table 3: Component Analysis. The four classes represent the novel classes in split 0. The final column, Avg, shows the average mIoU for novel classes across four split in the SemanticPOSS.

| Method | | | | Split 0 | | | | | Overall |
|---|---|---|---|---|---|---|---|---|---|
| Baseline | CRP | CRG | GCPL | Building | Car | Ground | Plants | Avg | Avg |
| ✓ | | | | 58.6 | 6.0 | 44.9 | 43.7 | 38.3 | 24.7 |
| ✓ | ✓ | | | 52.4 | 7.3 | 54.5 | 52.3 | 41.6 | 25.9 |
| ✓ | | ✓ | | 56.3 | 2.6 | 73.2 | 49.8 | 45.5 | 28.8 |
| ✓ | ✓ | ✓ | | 56.5 | 7.1 | 73.1 | 52.9 | 47.4 | 30.1 |
| ✓ | ✓ | ✓ | ✓ | **60.4** | **8.3** | **80.8** | **55.3** | **51.2** | **32.5** |

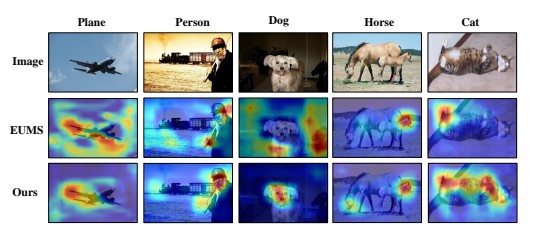

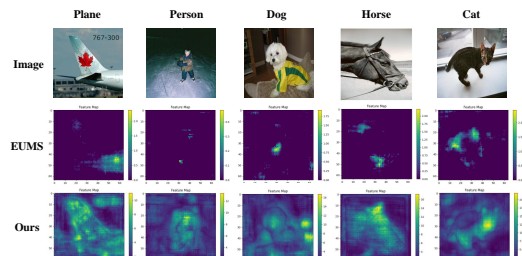

Figure 5: Comparison of novel class Grad-CAM visualization on the PASCAL-5$^i$ dataset.

Figure 6: Comparison of novel class feature map visualization on the COCO-20$^i$ dataset.

nodes, further improving performance. The fourth row combines CRP and CRG, enhancing causal representation capture and causal relationship modeling. Here we generate pseudo-labels for novel classes using Sinkhorn-Knopp Cuturi (2013), consistent with existing methods. The last row uses GCN to generate pseudo-labels for novel classes, boosting novel class segmentation. A visualization analysis is provided in Figure 4.

### 4.4 Extension for 2D NCD Semantic Segmentation

**Comparison with the SOTA method.** We extend our method to 2D NCD semantic segmentation. The current SOTA method is EUMS Zhao et al. (2021), which lacks the ability to capture causal representations and model the relationship between the base-novel classes as effectively as ours. To ensure fairness, we follow the same dataset partitioning strategy and experimental setup. We conducted experiments on the PASCAL-5$^i$ Shaban et al. (2017) and COCO-20$^i$ Nguyen and Todorovic (2019) datasets. As shown in the Table 4 and Table 5, our method outperforms EUMS in most fold, showing that our causal representation and reasoning approach is equally effective in 2D NCD.

Table 4: Performance on PASCAL-5$^i$ dataset.

| Method | PASCAL-5$^i$ | | | | |
|---|---|---|---|---|---|
| | Fold0 | Fold1 | Fold2 | Fold3 | Avg |
| EUMS | 69.8 | 60.1 | 56.3 | 50.2 | 59.1 |
| Ours | **71.9** | **62.5** | **57.3** | **53.1** | **61.2** |

Table 5: Performance on COCO-20$^i$ dataset.

| Method | COCO-20$^i$ | | | | |
|---|---|---|---|---|---|
| | Fold0 | Fold1 | Fold2 | Fold3 | Avg |
| EUMS | 42.39 | **26.89** | 19.75 | 18.19 | 26.81 |
| Ours | **43.23** | 25.91 | **20.30** | **18.56** | **27.00** |

**Visualization Analysis.** Figure 5 presents the Grad-CAM Selvaraju et al. (2016) visualization results. EUMS generates scattered feature maps, highlighting vague regions of interest that may miss important causal features. In contrast, our method produces more focused and clearer feature maps, effectively highlighting key causal regions and demonstrating superior causal reasoning capabilities. Figure 6 illustrates that EUMS results in sparse and dispersed activation regions, failing to capture key causal features. Our approach, however, through causal representation and reasoning, generates more concentrated and clear activations, better identifying causal relationships and significantly improving feature localization and visual accuracy.

## 5 Conclusion

We introduce a SCM to re-formalize the 3D-NCD problem and propose a novel approach called joint learning of causal representation and reasoning. Specifically, we first analyze the hidden confounding factors in the base class representations and the causal relationships between the base and novel classes through SCM. Based on this, we propose a causal representation prototype that captures the causal representation of the base class by eliminating hidden confounding factors. Then, we use a graph to model the causal relationship between the base class causal representation prototype and the novel class prototype, enabling causal reasoning from the base to the novel. Extensive experiments results on 3D and 2D NCD semantic segmentation demonstrate the superiorities of our method.

**Acknowledgment.** This work is supported by the National Nature Science Foundation of China (Nos. 62376186, 62472333).

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
