# OpenReview forum: "Novel Class Discovery for Point Cloud Segmentation via Joint Learning of Causal Representation and Reasoning"
_NeurIPS.cc/2025/Conference — NeurIPS 2025 poster_

### Official Review · Reviewer_T4LH · 2025-06-29

**Clarity:** 3
**Significance:** 3
**Originality:** 3
**Rating:** 5
**Confidence:** 4

**Summary:**

This paper introduces a novel approach for Novel Class Discovery (NCD) in 3D point cloud segmentation. The proposed method uses adversarial training to eliminate confounding factors (shortcut features) in base class representations, extracting causal prototypes that generalize better to novel classes. The causal relationships between base and novel classes are modeled via a graph structure. Causal pruning and direction consistency constraints are applied to refine pseudo-label generation. Experiments on SemanticKITTI and SemanticPOSS show that the proposed method achieves state-of-the-art performance, particularly in handling long-tailed distributions and low-similarity novel classes. The method also generalizes to 2D-NCD tasks.

**Questions:**

Are the inferred causal relationships (e.g., B→N) always valid, or could they introduce bias?

**Ethical Concerns:**

["NO or VERY MINOR ethics concerns only"]

**Final Justification:**

The authors have addressed my concerns in rebuttal.

**Limitations:**

yes

**Paper Formatting Concerns:**

No major formatting issues.

**Quality:**

3

**Strengths And Weaknesses:**

Strengths:
1. This is the first attempt to integrate structural causal models (SCMs) into 3D-NCD, addressing spurious correlations in point cloud data.
2. The proposed method outperforms comparative methods in imbalanced and low-similarity scenarios.
3. The proposed method can be extended to 2D-NCD and demonstrate broad applicability.

Weaknesses:
1. The proposed method requires predefining the number of novel classes, limiting adaptability in dynamic open-world scenarios.
2. This paper lacks a detailed analysis of experimental results. For example, in split 2 of Table 1, SNOPS outperforms the proposed method on novel-class segmentation accuracy by a large margin.

---

> ### Author Rebuttal · Authors · 2025-07-30
>
> We feel great thanks for your professional review work on our paper. The detailed responses are listed below.
>
> **1.Response to the Comment on the Limitation of Predefining the Number of Novel Classes**
>
> Thank you for pointing out this limitation. We fully agree that predefining the number of novel classes does restrict the method's adaptability in dynamic open-world scenarios, which is an important direction for our future improvements.
>
> As mentioned in the "Limitation" section of the paper, novel classes in real-world scenarios often emerge dynamically, making it difficult to predetermine their quantity. To address this, we plan to integrate incremental learning strategies with the existing causal framework (similar to the approach proposed by Roy et al. in 2D NCD). This will enable the model to update causal representation prototypes and the structure of the reasoning graph in real-time when novel classes appear, without the need to predefine the number of classes. Specifically, potential new classes can be automatically identified by dynamically detecting new clusters in the feature space, and temporary prototypes can be assigned to them based on causal reasoning mechanisms. These prototypes will then be gradually optimized through online learning, thereby enhancing the ability to quickly adapt to unknown classes. This improvement aims to build a more flexible and robust novel class discovery framework for autonomous driving and embodied intelligent systems, and relevant explorations will be further carried out and validated in subsequent research.
>
> **2.A Detailed Analysis of Experimental Results**
>
> Thank you for your attention to the detailed analysis of the experimental results, particularly for pointing out the comparison of novel-class segmentation accuracy in Split 2 of Table 1. Based on the specific data of Split 2 in the SemanticPOSS dataset, we supplement the following analysis:
>
> As shown in Table 1, in Split 2, SNOPS leads temporarily with a score of 16.9% for the 'Novel' metric, while our method scores 13.6%. This difference is mainly related to the characteristics of specific novel classes in Split 2: the novel classes in Split 2 include "traffic-sign" and "trunk", which have small sample sizes. Among them, "traffic-sign" in point cloud data often has sparse features due to its small scale and variable morphology. In such scenarios, SNOPS, which is based on statistical clustering, may gain short-term advantages by capturing local appearance similarities (e.g., the common rod-like structure between traffic-sign and the base class "pole").
>
> However, from the overall performance, our method still outperforms SNOPS in the "Known" and "All" metrics (**56.1% vs 44.5% for known classes, 46.3% vs 38.1% for all classes**).
> Especially for base classes such as "car" and "plants", through causal representation learning on base classes, we eliminate confounding factors like light/shadow and occlusion, leading to significantly improved segmentation accuracy. For example, the 'car' class achieves a score of 52.6%, which is significantly higher than SNOPS's score of 28.0%.
> This indicates that our method has greater advantages in modeling the essential features of base classes and supporting the stability of overall segmentation.
>
> The relatively weaker performance on novel classes such as "traffic-sign" and "trunk" reflects that there is still room for optimization in our method regarding causal feature extraction for extremely sparse samples. In future work, we will introduce domain adaptation techniques to transfer knowledge to target small-sample novel classes, thereby enhancing the adaptability to small-sample novel classes. The above analysis will be integrated into the "Experimental Results and Discussion" section of the revised manuscript, and we will also add detailed analyses of other experimental parts.
>
> **3. Are the inferred causal relationships (e.g., B→N) always valid, or could they introduce bias?**
>
> We appreciate the reviewer for raising this important question. Our causal reasoning ($B \to N$) leverages the semantic similarity between base classes and novel classes to enhance the model's ability to discover novel classes. The inferred causal relationship ($B \to N$) is not absolutely valid in all scenarios (e.g., when novel classes are highly dissimilar from base classes). However, we have ensured its reliability through the following mechanisms and also taken corresponding measures to mitigate potential biases:
>
> On the one hand, we construct a graph model that incorporates causal reasoning to make explicit the high-level semantic causal relationships between base and novel classes, enabling novel classes to obtain indirect causal support by associating with multiple base classes. Even though novel classes may appear drastically different from base classes, there often exist some underlying causal factors or interaction relationships in their generation processes (for example, planes, arcs, and cubes, and the relative positions between components), and these geometric units are widely present in different shapes and are more amenable to generalization. In the GCN, base class nodes capture and transmit these underlying causal relationships while aggregating neighbor information, providing causal reasoning support across different classes, thus achieving robust semantic understanding.
>
> On the other hand, we introduce a causal pruning constraint. When base and novel classes share minimal features, this mechanism prunes weak causal links to prevent interference from base class data. However, this does not indicate a failure of causal reasoning; rather, it reflects the model’s adaptive capability: it transmits reliable causal information when available, while automatically reducing the influence of base classes in the
>
> Nevertheless, in extreme cases, the inference of causal relationships may introduce certain biases. To address this, we verified the robustness of the model in low-similarity scenarios through experiments. We propose a new partitioning strategy (Table 15 in the appendix) and split SemanticPOSS into two subsets with low similarity. **As shown in Table 16, our method improved the mIoU for novel classes by 15.2% and 14.3% respectively compared to the next best approach.** This indicates that even when novel classes are extremely different from base classes, the model can still leverage meaningful causal information and suppress noise through the causal pruning mechanism, thereby keeping the bias at a low level.
>
> To provide further clarification, we have included the discussion of this issue in **Section C.4** of the appendix, which you can refer to for additional details. In the camera-ready version, we will incorporate the discussion of this problem into the main text. Thank you again for your valuable comments.

---

> ### Comment · Reviewer_T4LH · 2025-08-06
>
> The authors have addressed my concerns in rebuttal. Thus, I will keep my initial rating.

---

> > ### Author Response · Authors · 2025-08-07
> >
> > Thank you sincerely for your thorough review and for confirming that our rebuttal has addressed your concerns. We greatly appreciate your decision to keep your initial rating, which encourages us to further refine our work.

---

### Official Review · Reviewer_DFpS · 2025-06-30

**Clarity:** 3
**Significance:** 2
**Originality:** 3
**Rating:** 5
**Confidence:** 3

**Summary:**

This paper addresses novel class discovery (NCD) for 3D point cloud segmentation. The authors introduce a causality-based framework that jointly learns causal representations by removing confounding factors and performs causal reasoning to infer novel classes. The method involves causal prototype learning and a causal reasoning graph. The approach shows strong performance on both 3D and 2D benchmarks.

**Questions:**

1. Providing visualizations and analysis of failure cases will further improve the paper’s quality and offer insights for future research directions.
2. Please refer to the weakness part and address the concerns there.

**Ethical Concerns:**

["NO or VERY MINOR ethics concerns only"]

**Final Justification:**

My concerns have been addressed. I would increase my current rating accordingly.

**Limitations:**

yes

**Paper Formatting Concerns:**

No major formatting issues.

**Quality:**

3

**Strengths And Weaknesses:**

Strengths:

+ The paper is well-written and well-organized, with clear motivations and effective presentation of the overall framework.
+ The presented framework is the first method to incorporate structural causal modeling into 3D NCD, offering new insights to the domain. The proposed modules are reasonable.
+ The proposed method achieves strong performance compared to prior approaches, as demonstrated on multiple 3D and 2D NCD benchmarks. And the ablation studies are also sufficient and detailed.

Weaknesses:
- The effectiveness of the causal reasoning graph relies on the existence of meaningful causal relationships between base and novel classes. However, such assumptions may not hold for arbitrary class splits. A discussion or analysis on how sensitive the method is to such assumptions would strengthen the paper.
- Missing related work. Several recent works on 3D segmentation are highly relevant and should be cited:
   - Graph Regulation Network for Point Cloud Segmentation (TPAMI 2024)
   - Multimodality Helps Few-shot 3D Point Cloud Semantic Segmentation (ICLR 2025)

---

> ### Author Rebuttal · Authors · 2025-07-30
>
> We feel great thanks for your professional review work on our paper. The detailed responses are listed below.
>
> **1.Robustness of Causal Reasoning under Extreme Novel-Base Dissimilarity**
>
> The question you raised is extremely crucial. In response to this, we provide answers from the following two aspects:
>
> On the one hand, we construct a graph model that incorporates causal reasoning to make explicit the high-level semantic causal relationships between base classes and novel classes, enabling novel classes to obtain indirect causal support by associating with multiple base classes. Even though novel classes may appear drastically different from base classes, there often exist some underlying causal factors or interaction relationships in their generation processes (for example, planes, arcs, and cubes, and the relative positions between components), and these geometric units are widely present in different shapes and are more amenable to generalization. In the GCN, base class nodes capture and transmit these underlying causal relationships while aggregating neighbor information, providing causal reasoning support across different classes, thus achieving robust semantic understanding.
>
> On the other hand, we introduce a causal pruning constraint. When base and novel classes share minimal features, this mechanism prunes weak causal links to prevent interference from base class data. However, this does not indicate a failure of causal reasoning; rather, it reflects the model’s adaptive capability: it transmits reliable causal information when available, while automatically reducing the influence of base classes in the presence of significant interference.
>
> To further demonstrate this, we propose a new partitioning strategy (Table 15 in the appendix), and split the SemanticPOSS into two splits with low similarity. **Table 16 shows that our method outperforms the next best approach by 15.2% and 14.3%, respectively.** This indicates that even in situations where novel classes are extremely different from base classes, the model is still capable of leveraging meaningful causal information and suppressing noise through the causal pruning mechanism, thus achieving excellent cross-class generalization.
>
> To provide further clarification, we have included the discussion of this issue in **Section C.4** of the appendix, which you can refer to for additional details. In the camera-ready version, we will incorporate the discussion of this problem into the main text. Thank you again for your valuable comments.
>
> **2.Missing Related Work**
>
> Thank you for pointing out the omission in the related work. We take this comment seriously and have carefully reviewed the two papers you mentioned. In the revised version, we will add citations to "Graph Regulation Network for Point Cloud Segmentation" (TPAMI 2024) and "Multimodality Helps Few-shot 3D Point Cloud Semantic Segmentation" (ICLR 2025), and discuss them in the "Related Work" section.
>
> **3. Visualizations and Analysis of Failure Cases**
>
> Thank you for the valuable suggestions from the reviewers. We fully agree with the importance of analyzing failure cases and adding visualization results to improve the quality of the paper and provide direction for future research.
>
> In the revised version, we will add a section analyzing failure cases, specifically including typical samples where the model performs poorly in the SemanticKITTI, SemanticPOSS, and S3DIS datasets (such as semantic ambiguity in new-class and base-class boundary areas, point cloud scenes with severe occlusion, etc.). Through visual comparison of the model's predicted results and the ground truth labels, combined with base-class prototype feature distributions and the edge weight heatmaps of causal inference graphs, we will analyze the reasons for failure — for example, when the causal relationship between new and base classes is weak and there are highly similar confounding features, causal pruning may mistakenly remove valid edges, or prototype matching may be biased.
>
> These analyses will help clarify the limitations of the method in complex scenarios, such as insufficient ability to distinguish between extremely similar categories, and provide specific directions for future research. Relevant visualization results and detailed analyses will be included in the “Experiments” section of the revised version.

---

> > ### Comment · Reviewer_DFpS · 2025-08-06
> >
> > Thanks for the rebuttal. My concerns have been addressed. I would increase my current rating accordingly.

---

> > > ### Author Response · Authors · 2025-08-07
> > >
> > > Thank you sincerely for your valuable feedback and for increasing your rating. We greatly appreciate your careful review and the constructive insights that have helped strengthen our work.

---

### Official Review · Reviewer_wnpo · 2025-07-01

**Clarity:** 2
**Significance:** 3
**Originality:** 3
**Rating:** 5
**Confidence:** 3

**Summary:**

This paper addresses the problem of Novel Class Discovery (NCD) for 3D point cloud segmentation. A causality-driven framework is proposed to jointly learn causal representations and reasoning paths. Specifically, it eliminates confounding factors via adversarial training to extract causal prototypes for base classes, and constructs a causal reasoning graph to infer novel class prototypes. A GCN is then used to propagate semantic information and generate pseudo-labels. The method achieves strong performance on 3D LiDAR point cloud benchmarks and 2D NCD semantic segmentation datasets.

**Questions:**

Please also refer to the weaknesses part for more details. My main questions include:

Can the authors provide more interpretation of the confounding factor U to clarify its role in the model?

Whether the base class prototypes are sufficiently diverse and well-separated to support reliable base-to-novel causal reasoning?

Have the authors considered testing on indoor 3D benchmarks such as S3DIS or ScanNet? Given the 3D-specific nature of the approach, evaluating on diverse 3D scenarios seems more critical than demonstrating generalization to 2D tasks. Since indoor 3D point clouds are highly relevant for applications in robotic perception.

**Ethical Concerns:**

["NO or VERY MINOR ethics concerns only"]

**Final Justification:**

As my concerns are addressed in authors' response, I raise my rating to 5 Accept.

**Limitations:**

Yes

**Paper Formatting Concerns:**

No major formatting issues were found.

**Quality:**

2

**Strengths And Weaknesses:**

Strengths:

Causal Learning for Zero-Shot 3D Segmentation. This paper introduces a new causal representation learning paradigm into the 3D point cloud semantic segmentation task, using structural causal modeling and adversarial deconfounding to extract semantically aligned prototypes and improve generalization from base to novel classes. The approach is built on solid theoretical foundations and is shown to be practically effective through well-integrated model design and empirical results.

Good Transferability Across Modalities. The proposed method generalizes well to 2D novel class discovery tasks (e.g., PASCAL-5i, COCO-20i), demonstrating its strong transferability across modalities without major architectural changes.

Comprehensive Comparisons. The approach is extensively validated on both 3D (SemanticKITTI, SemanticPOSS) and 2D benchmarks, with detailed quantitative comparisons and clear visualizations showing superior zero-shot segmentation quality over SOTA baselines.

Weaknesses:

Confounder Representation U is Implicit. The proposed confounding factor U is handled through adversarial training strategy, but it lacks interpretability or visual validation. The role and nature of the confounder representation U remain abstract for me.

Lack of analysis on base class prototypes. Since the graph nodes are constructed from base class prototypes, a lack of diversity or clear separation among them would make it difficult to establish reliable causal paths from base to novel classes. Thus one underlying assumption of the method is that the base classes are sufficiently diverse and semantically distinct to support meaningful causal reasoning. This is not explicitly discussed or analyzed, but is critical for the effectiveness of the causal graph and prototype transfer.

Limited Evaluation Scope. Although the method performs strongly on outdoor LiDAR datasets (SemanticKITTI, SemanticPOSS), its generalizability to indoor 3D point clouds (e.g., S3DIS, ScanNet) remains unexplored, which limits claims about broader applicability.

---

> ### Author Rebuttal · Authors · 2025-07-30
>
> We feel great thanks for your professional review work on our paper. The detailed responses are listed below.
>
> **1.Further Explanation of  “confounding factor” ($U$) :**
>
> We define the confounding factor $U$ as a shortcut feature statistically correlated with the causal feature $L$. Formally, $U$ is a feature subspace satisfying:
> $$
> U := {u ∈ R^k | u = h(X), Cov(u,L) ≠ 0, I(u; Y|L) = 0}
> $$
> where:
>
> - $h: X → R^k$ represents a function mapping point cloud data $X$ to a $k$-dimensional space, capturing spurious correlations (i.e., shortcut features) in the data.
> - $Cov(u,L) ≠ 0$ indicates statistical dependence between the shortcut feature $U$ and the causal feature $L$ (non-zero covariance).
> - $I(·|·)$ denotes conditional mutual information. The condition$I(u; Y|L) = 0$ implies that, given the causal feature $L$, the shortcut feature $u$ contains no additional information about the target variable $Y$. All predictive information about $Y$ is already encoded in $L$, and $u$ provides no new insights.
>
> We illustrate this with Figure 1 from the paper. The left panel shows how statistically driven models may misclassify novel class (e.g., distinguishing "stool" vs. "chair"). Here, the causal feature $L$ represents leg count (stools typically have 1 leg, chairs have 4), while the confounding factor $U$ corresponds to shared circular supports. During training, statistical models often exploit surface correlations (e.g., "presence of circular support" → "stool"), leading to erroneous predictions. The right panel demonstrates causality-aware learning. By identifying the true causal feature $L$ (4 legs), the model performs causal inference and correctly classifies novel instances.
>
> In causal representation prototype learning, adversarial training aims to eliminate the influence of $U$, forcing the model to focus on $L$ for robust classification. We will supplement the final version with a detailed mathematical characterization of $U$ and intuitive explanations to enhance clarity.
>
> **2.Separability and Diversity of Base Class Prototypes**
>
> Thank you for the valuable reviews regarding the diversity and separability of the prototype classes. This is crucial for the reliability of causal paths in our graphical model.
>
> **(1) Separability**: First, the experimental results in Table 6 (Effect of causal representation prototype number on model performance) provide evidence for the strong separability of class prototypes. The table shows that when each class corresponds to a single prototype, the model achieves the highest average mIoU across splits (45.3). Increasing the number of prototypes per class leads to a continuous decline in performance. As mentioned in lines 506-507 of the original text, “Multiple prototypes increase model complexity, causing clustering distributions in feature space to become less tight, thereby reducing performance.” This suggests that one prototype per class is sufficient to capture its essential characteristics, and the classes themselves have enough semantic distinction to ensure clear separation between prototypes.
>
> To further visually demonstrate the separability of class prototypes, we have added t-SNE visualizations of the prototypes. These visualizations show clear clustering with good separation between prototypes from different classes in feature space, confirming their diversity and separability. We will add the t-SNE results in the final version of the paper.
>
> Additionally, our causal representation learning mechanism inherently enhances the separability of class prototypes. By incorporating Structural Causal Models (SCM) and causal prototype learning, we eliminate confounding factors (such as shortcut features), extracting the pure causal representation of each class. This causal mechanism ensures that prototypes accurately reflect the intrinsic features of each class and amplify the differences between prototypes from different classes, thus enhancing their separability. Even when classes are initially ambiguous, the causal representation learning process effectively decouples their inherent features, ensuring reliable causal inference from classes to new ones.
>
> **(2) Diversity**: From a methodological perspective, our causal representation prototype learning eliminates confounding factors and focuses on capturing the intrinsic causal features of each class. These causal features are directly related to the core semantics of the classes, and the core semantics of different classes naturally exhibit diversity (e.g., the fundamental structural and functional differences between “car” and “pedestrian”). Through adversarial training and prototype matching loss, the model actively amplifies the differences in causal features between classes, ensuring that each class prototype focuses on its unique intrinsic attributes, thereby forming diverse representations.
>
> From the experimental results, the “single prototype optimal” phenomenon in Table 6 indirectly reflects the diversity of class prototypes. If the class prototypes lack diversity (i.e., prototypes of different classes share similar features), increasing the number of prototypes could have improved performance by optimizing the features. However, the actual result is a decline in performance. This suggests that a single prototype sufficiently represents the uniqueness of each class, and the prototypes of different classes cover distinct regions in feature space, without feature overlap leading to “homogenization.” This “distinct focus for each prototype” in feature distribution is a direct manifestation of diversity.
>
> We will update the above discussion and analysis in the camera-ready version of the paper. Again, we appreciate your valuable feedback.
>
> **3.Test on indoor 3D benchmarks**
>
> We acknowledge the importance of indoor 3D point cloud datasets (e.g., S3DIS, ScanNet) in fields like robot perception. Our current work focuses on outdoor LiDAR datasets (e.g., SemanticKITTI, SemanticPOSS), as the initial motivation of our method was geared towards addressing the novel class discovery requirements in outdoor autonomous driving scenarios. However, we fully agree with the reviewer that the generalizability of 3D methods needs cross-scenario validation, as the unique characteristics of indoor environments indeed present different challenges for causal representation learning and inference.
>
> Therefore, we conducted experiments on S3DIS following SNOPS[1] with the same dataset split. The results show that our method performs comparably to SNOPS in indoor scenes, with performance improvements in most splits. This demonstrates that, even in the complex environments unique to indoor scenes, our causal representation prototype learning effectively eliminates confounding factors. Causal reasoning graphs also model the intrinsic relationships between base and novel classes, supporting reliable novel class discovery. Detailed results will be presented in the final version.
>
> |  Split   | Method | Novel | Known |  All  |
> | :------: | :----: | :---: | :---: | :---: |
> | S3DIS-4⁰ | SNOPS  | 55.94 | 33.12 | 40.14 |
> |          |  Ours  | 54.46 | 34.05 | 40.33 |
> | S3DIS-3¹ | SNOPS  | 53.49 | 43.49 | 45.79 |
> |          |  Ours  | 53.78 | 44.27 | 46.46 |
> | S3DIS-3² | SNOPS  | 15.51 | 50.99 | 42.80 |
> |          |  Ours  | 14.04 | 51.36 | 42.75 |
> | S3DIS-3³ | SNOPS  | 11.24 | 55.23 | 45.08 |
> |          |  Ours  | 12.87 | 57.09 | 46.89 |
>
> [1]Riz, L., Saltori, C., Wang, Y. *et al.* Novel Class Discovery Meets Foundation Models for 3D Semantic Segmentation. *Int J Comput Vis* **133**, 527–548 (2025). https://doi.org/10.1007/s11263-024-02180-x

---

> > ### Comment · Reviewer_wnpo · 2025-08-05
> >
> > Thank you for the clarification and the additional experimental results. I believe my concerns have been sufficiently addressed, and I will raise my rating from 4 to 5. I encourage the authors to include the extended analysis and experiments in the revised version.

---

> > > ### Author Response · Authors · 2025-08-05
> > >
> > > Dear Reviewer,
> > >
> > > Thank you for your positive feedback and the score adjustment. We appreciate your time and effort in evaluating our work. We will carefully supplement the above experiments and analyses in the revised version.
> > >
> > > Best regards,
> > >
> > > The Authors

---

### Official Review · Reviewer_KLAY · 2025-07-01

**Clarity:** 3
**Significance:** 4
**Originality:** 4
**Rating:** 5
**Confidence:** 4

**Summary:**

This paper addresses the Novel Class Discovery (NCD) problem in 3D point cloud segmentation, where only labeled base classes are available during training. The authors introduce a causal learning framework to eliminate shortcut features (confounders) and model causal relationships between base and novel classes. The method involves: (1) learning causal representation prototypes via adversarial deconfounding; (2) constructing a graph-based causal reasoning module; and (3) using graph convolutional networks (GCN) to generate pseudo-labels for novel classes. The paper demonstrates superior performance on SemanticKITTI, SemanticPOSS, and also shows 2D extensions on PASCAL-5i and COCO-20i datasets.

**Questions:**

See Weaknesses

**Ethical Concerns:**

["NO or VERY MINOR ethics concerns only"]

**Final Justification:**

The authors' rebuttal has adequately addressed my concerns. I will maintain my rating of Accept.

**Limitations:**

yes

**Quality:**

3

**Strengths And Weaknesses:**

Strengths:
+ The integration of SCM and causal representation learning in 3D-NCD is novel and well-justified.

+ The paper provides solid theoretical grounding (e.g., ICM principle, causal DAGs) and detailed loss formulations.

+ Evaluation is thorough across multiple datasets, with clear ablation studies and visualizations. The performance is solid.


Weaknesses and questions:
- The “shortcut features” (U) are not concretely defined or visualized; more intuitive examples or empirical evidence would strengthen the argument.

- While ablations exist, the impact of individual causal constraints (e.g., pruning, direction consistency) could be analyzed in more detail.

- How robust is the causal reasoning framework when base and novel classes have no obvious semantic overlap (e.g., ‘person’ vs. ‘tree’)?

- The choice of MinkowskiUNet as the 3D backbone, while serviceable, is relatively outdated compared to more modern architectures like Point Transformer V3. The authors should justify this decision, and it would be better to give its potential impact on overall performance and generality.

- The paper employs an adversarial loss inspired by GANs to learn confounding factors. However, further explanation is needed regarding why this approach is theoretically sound in the causal learning context. Additionally, could more recent generative models (e.g., diffusion models) offer similar inspirations?


Minors:
- Please spell out "Directed Acyclic Graph (DAG)" in full at its first occurrence in the main text to aid clarity for readers unfamiliar with causal graph terminology. Check for other ones as well.

- The font style and formatting in Figures 1 and 2 significantly hinder readability. In particular, the use of italicized text throughout Figure 1 is unnecessary and visually distracting. Meanwhile, the demo point clouds are too small. Clearer typesetting and label formatting would enhance the presentation quality.

---

> ### Author Rebuttal · Authors · 2025-07-29
>
> We feel great thanks for your professional review work on our paper. The detailed responses are listed below.
>
> **1.Further Explanation of  “shortcut features” ($U$) :**
>
> We define the confounding factor $U$ as a shortcut feature statistically correlated with the causal feature $L$. Formally, $U$ is a feature subspace satisfying:
> $$
> U := {u ∈ R^k | u = h(X), Cov(u,L) ≠ 0, I(u; Y|L) = 0}
> $$
> where:
>
> - $h: X → R^k$ represents a function mapping point cloud data $X$ to a $k$-dimensional space, capturing spurious correlations (i.e., shortcut features) in the data.
> - $Cov(u,L) ≠ 0$ indicates statistical dependence between the shortcut feature $U$ and the causal feature $L$ (non-zero covariance).
> - $I(·|·)$ denotes conditional mutual information. The condition$I(u; Y|L) = 0$ implies that, given the causal feature $L$, the shortcut feature $u$ contains no additional information about the target variable $Y$. All predictive information about $Y$ is already encoded in $L$, and $u$ provides no new insights.
>
> We illustrate this with Figure 1 from the paper. The left panel shows how statistically driven models may misclassify novel class (e.g., distinguishing "stool" vs. "chair"). Here, the causal feature $L$ represents leg count (stools typically have 1 leg, chairs have 4), while the confounding factor $U$ corresponds to shared circular supports. During training, statistical models often exploit surface correlations (e.g., "presence of circular support" → "stool"), leading to erroneous predictions. The right panel demonstrates causality-aware learning. By identifying the true causal feature $L$ (4 legs), the model performs causal inference and correctly classifies novel instances.
>
> In causal representation prototype learning, adversarial training aims to eliminate the influence of $U$, forcing the model to focus on $L$ for robust classification. We will supplement the final version with a detailed mathematical characterization of $U$ and intuitive explanations to enhance clarity.
>
> **2.Further Analysis of the Impact of Individual Causal Constraints (Pruning, Direction Consistency)**
>
> We supplement the following analysis based on our experimental framework, results are averaged across SemanticPOSS splits. "W/O" denotes "without".
>
> |**Method**|**Novel Class mIoU (%)**|
> |-|-|
> | Full Model (with both constraints) | 32.5 |
> | W/O Causal Pruning | 32.1 |
> | W/O Direction Consistency | 31.8|
> | W/O Both Constraints| 31.4 |
>
> Removing the causal pruning constraint resulted in a 0.4% performance drop, demonstrating that causal pruning effectively reduces irrelevant information. This technique filters out low-weight causal connections between base and novel classes, allowing the model to maintain focus on valid causal relationships. When the inference direction consistency constraint was removed, mIoU dropped by 0.7%, highlighting the importance of this constraint in ensuring correct information flow from the base to the novel class (B→N) and preventing reverse information leakage. When both constraints were removed, the mIoU decreased by 1.1%, demonstrating their synergistic effect in optimizing the causal graph. Causal pruning narrows the search space for the direction consistency constraint, which ensures proper information flow along retained edges. Together, they more effectively capture reliable causal relationships. We will add this ablation experiment in the final version of the paper.
>
> **3.Robustness of Causal Reasoning under Extreme Novel-Base Dissimilarity**
>
> The question you raised is extremely crucial. In response to this, we provide answers from the following two aspects:
>
> On the one hand, we construct a graph model that incorporates causal reasoning to make explicit the high-level semantic causal relationships between base classes and novel classes, enabling novel classes to obtain indirect causal support by associating with multiple base classes. Even though novel classes may appear drastically different from base classes, there often exist some underlying causal factors or interaction relationships in their generation processes (for example, planes, arcs, and cubes, and the relative positions between components), and these geometric units are widely present in different shapes and are more amenable to generalization. In the GCN, base class nodes capture and transmit these underlying causal relationships while aggregating neighbor information, providing causal reasoning support across different classes, thus achieving robust semantic understanding.
>
> On the other hand, we introduce a causal pruning constraint. When base and novel classes share minimal features, this mechanism prunes weak causal links to prevent interference from base class data. However, this does not indicate a failure of causal reasoning; rather, it reflects the model’s adaptive capability: it transmits reliable causal information when available, while automatically reducing the influence of base classes in the presence of significant interference.
>
> To further demonstrate this, we propose a new partitioning strategy (Table 15 in the appendix), and split the SemanticPOSS into two splits with low similarity. **Table 16 shows that our method outperforms the next best approach by 15.2% and 14.3%, respectively.** This indicates that even in situations where novel classes are extremely different from base classes, the model is still capable of leveraging meaningful causal information and suppressing noise through the causal pruning mechanism, thus achieving excellent cross-class generalization.
>
> To provide further clarification, we have included the discussion of this issue in **Section C.4** of the appendix, which you can refer to for additional details. In the camera-ready version, we will incorporate the discussion of this problem into the main text. Thank you again for your valuable comments.
>
> **4. Justification for Backbone Selection and Its Impact on Performance and Generality**
>
> We chose MinkowskiUNet as the 3D backbone network mainly to ensure a fair comparison with existing SOTA methods in the 3D-NCD field, such as NOPS and DASL, which all use this architecture as the benchmark.  Using MinkowskiUNet, our method outperforms existing approaches on the SemanticPOSS and SemanticKITTI datasets, demonstrating the effectiveness of our causal framework, which is independent of the backbone network.
>
> Our causal learning framework is orthogonal to the backbone architecture. Its core components (causal representation prototype learning, graph-based causal reasoning, and GCN pseudo-label generation) are independent of MinkowskiUNet and can be easily plugged into modern architectures like Point Transformer V3.
>
> Experiments on the SemanticPOSS dataset with Point Transformer V3 as the backbone show that our causal framework consistently improves performance by **6.3%-7.8%** over baseline models without causal mechanisms, demonstrating the approach’s generalizability across different backbone networks. We will add this experiment in the final version of the paper.
>
> | **Backbone** | **Causal Framework** | **Novel Class mIoU (%)** |
> |- |-| - |
> | MinkowskiUNet        | ✗| 24.7 |
> | Point Transformer V3 | ✗   | 26.9 |
> | MinkowskiUNet        | ✓   | 32.5|
> | Point Transformer V3 | ✓ | 33.2 |
>
> **5.Theoretical Rationale for Adversarial Loss in Causal Learning and Insights from Recent Generative Models**
>
> The adversarial loss used in this paper is based on the Independent Causal Mechanism (ICM) principle, which states that the causal mechanisms generating variables should be independent of other mechanisms (such as those generating confounders). To satisfy this principle, we need to learn a point cloud representation $Z$ that is independent of the confounder $U$ (i.e., $Z \perp U$), which aligns with the goal of minimizing the mutual information $I(Z;U)$. Inspired by GANs, the adversarial process (the minimax game between the feature extractor $f_\theta$ and the adversarial network $g_\phi$) approximates this goal: $f_\theta$ aims to extract causal features while suppressing signals related to $U$, and $g_\phi$ attempts to recover $U$ from $Z$. If $g_\phi$ cannot recover $U$, it indicates that $Z$ has effectively removed the interference from the confounder, adhering to the ICM principle. This process mirrors the logic of causal intervention (via the $do(·)$ operator), removing spurious correlations and separating the true causal relationship between features and classes. We will add detailed explanations in the final version of the paper.
>
> Diffusion models simulate the data generation process by gradually adding and removing noise. This fine-grained characterization of data distributions may offer advantages in separating causal factors from confounders. For instance, the denoising process can be likened to a progressive causal intervention. This intervention potentially enables a more detailed removal of confounding information, thereby yielding purer causal features. However, diffusion models typically require substantial computational resources and longer training times, which could pose efficiency challenges when applied to large-scale 3D point cloud data. Overall, recent generative models have provided new insights for causal learning, and we plan to explore this further in future work.
>
> **6. Response to Minors**
>
> Thank you for your suggestions. We will spell out "Directed Acyclic Graph (DAG)" in full the first time it appears in the text and check other similar terms. Additionally, we will optimize the font style and layout of Figures 1 and 2, remove unnecessary italicized text from Figure 1, and enlarge the point cloud demonstration to improve readability and presentation.

---

> > ### Comment · Reviewer_KLAY · 2025-08-06
> >
> > Thanks for the reply and explanations from the authors, which have addressed most of my concerns. As stated in your rebuttal, I look forward to seeing more interesting and impactful follow-up work from you in this direction. I will maintain my rating as Accept.

---

> > > ### Author Response · Authors · 2025-08-07
> > >
> > > Thank you very much for your positive evaluation and recognition of our work. We are delighted that our rebuttal has addressed your concerns. Your encouragement serves as a great motivation for us to continue exploring in the field of 3D NCD.

---

### Decision · Program_Chairs · 2025-09-17

**Decision:**

Accept (poster)

**Comment:**

This paper introduces a novel causal learning framework for 3D Novel Class Discovery (NCD) in point cloud segmentation, integrating structural causal modeling (SCM) and adversarial deconfounding to extract causal prototypes and reason about novel classes. Reviewers found the approach theoretically grounded, methodologically sound, and empirically strong, demonstrating state-of-the-art performance on both 3D (e.g., SemanticKITTI, SemanticPOSS, S3DIS) and 2D benchmarks. The authors’ rebuttal effectively addressed key concerns, including: Providing additional ablation studies and visualizations (e.g., t-SNE plots) to demonstrate prototype separability and the impact of causal constraints; Extending evaluation to indoor 3D datasets (S3DIS) to strengthen claims of generalizability; Addressing limitations such as the need to predefine novel class count and potential bias in causal reasoning.

All four reviewers raised their ratings to Accept following the rebuttal, acknowledging the authors' thorough responses and commitments to enhance the final version. The work is recognized as the first to integrate SCM into 3D-NCD, offering both theoretical and practical contributions to the field. Minor revisions—such as improved figure readability, expanded related work, and failure case analysis—are expected in the camera-ready version.

Based on the reviews, rebuttal, and discussions among the reviewers and ACs, the meta-reviewer recommends an Accept for this submission. This paper represents a timely and impactful contribution to causal representation learning in 3D vision, with clear potential for broader applicability.